# Healthy Obese Subjects Differ in Chronotype, Sleep Habits, and Adipose Tissue Fatty Acid Composition from Their Non-Healthy Counterparts

**DOI:** 10.3390/nu13010119

**Published:** 2020-12-31

**Authors:** Nathaly Torres-Castillo, Erika Martinez-Lopez, Barbara Vizmanos-Lamotte, Marta Garaulet

**Affiliations:** 1Department of Molecular Biology and Genomics, Institute for Translational Nutrigenetics and Nutrigenomics, Health Sciences University Center, University of Guadalajara, Guadalajara, Jalisco 44340, Mexico; nathaly.torrescas@academicos.udg.mx (N.T.-C.); erikamtz27@yahoo.com.mx (E.M.-L.); 2Department of Human Reproduction, Institute for Translational Nutrigenetics and Nutrigenomics, Health Sciences University Center, University of Guadalajara, Guadalajara, Jalisco 44340, Mexico; bvizmanos@yahoo.com.mx; 3Department of Physiology, University of Murcia, 30100 Murcia, Spain; 4Research Biomedical Institute of Murcia (IMIB)-Arrixaca, 30120 Murcia, Spain; 5Division of Sleep and Circadian Disorders, Brigham, and Women’s Hospital, Boston, MA 02115, USA

**Keywords:** metabolically benign obesity, adipose tissue, intra-abdominal fat, fatty acids, hormones, lifestyle, chronotype, case-control study, obesity

## Abstract

Obesity is not the same in all individuals and two different phenotypes have been described: metabolically healthy obesity (MHO) and metabolically unhealthy obesity (MUO). The aim of this study was to identify factors that explain metabolic health status in a rigorously matched Spanish population. Subcutaneous and visceral fat, adipocyte size and fatty acid composition, cardiometabolic markers in serum, and lifestyle habits were assessed. Higher physical activity in the mornings (Odds Ratio (95% Confidence Interval) (OR (95% CI) = 1.54 (1.09–2.18), *p* = 0.01)), earlier bedtimes (8:30–10:30 pm) (OR = 2.11 (1.02–4.36), *p* = 0.04), a complete breakfast (OR = 1.59 (1.07–2.36), *p* = 0.02), and a greater number of meals per day (4.10 ± 0.05 vs. 3.93 ± 0.05, *p* < 0.01), were associated with the MHO phenotype. Concentrations of 20:5 n-3 eicosapentaenoic acid (0.26 ± 0.46 vs. 0.10% ± 0.11%, *p* = 0.04) and 18:3 n-6 gamma-linolenic acid (0.37 ± 0.24 vs. 0.23% ± 0.22%, *p* = 0.04) in subcutaneous adipocytes were higher and omental adipocyte size (187 094 ± 224 059 µm^3^ vs. 490 953 ± 229 049 µm^3^, *p* = 0.02) was lower in MHO subjects than in those with MUO. Visceral fat area differed between MHO and MUO subjects (135 ± 60 cm^2^ vs. 178 ± 85 cm^2^, *p* = 0.04, respectively). The study highlights specific lifestyle habits that could form part of obesity therapies, not only involving healthier eating habits but also earlier sleeping and exercise patterns.

## 1. Introduction

Metabolically healthy obese (MHO) subjects, defined as those individuals who, despite excess body fat, remain free of metabolic abnormalities and increased cardiometabolic risk, usually follow specific lifestyle habits in their daily life [1,2]. For example, they are less sedentary during the day, take more physical activity, have a higher intake of fiber [3], follow a dietary pattern that includes fruits, vegetables and fish [4], and smoke less [5]. Nevertheless, limited data exist regarding other important lifestyle factors that are increasingly related to obesity, such as individual chronotype, the timing of food intake and the timing of sleep.

Recent studies suggest that chronodisruption—a disturbance of circadian rhythms—could have an effect at physiological, endocrinological, and behavioral levels [6]. In this sense, it has been demonstrated that meal timing influences glucose homeostasis [7] and late eaters have worse eating behaviors, such as skipping breakfast more frequently [8]. Besides, shorter sleepers were seen to have higher body fat [9] and a higher risk of developing metabolic alterations [10]. These observations highlight a link between lifestyle factors and physiological traits that are considered important for maintaining health. However, little is known regarding chronodisruption and MHO.

Besides the above, adipose tissue (AT) characteristics may also be involved in determining MHO. However, adipocyte size is difficult to assess, since it requires AT biopsies, so that very few studies use this concept to differentiate MHO and metabolically unhealthy obesity (MUO), especially in terms of visceral AT, which, furthermore, is difficult to access.

Moreover, fatty acids (FAs) are involved in inflammation through the production of eicosanoids and cytokines [11]. Previous studies have measured FA profiles in an attempt to differentiate MHO from MUO, but FAs were only analyzed in plasma [12] even though FA measurement at cellular level is also important since some may act as signaling molecules [13].

Although behavioral, physiological, morphological, and endocrinological characteristics are involved in MHO [14,15], no study has considered all these aspects together in an integrative way, and few studies match the subjects according to the obesity degree, which makes it difficult to interpret results and may explain inconsistencies among studies [1,2,14,15,16].

The current work contemplates new lifestyle factors that have been shown to be involved in obesity, such as the individual chronotype, eating behavior, including food timing, and sleep characteristics and timing. In addition, AT characteristics, such as adipocyte size and number, and the composition of FAs, together with hormonal concentrations, are considered, thus conforming an integrative approach to characterizing factors that favor the health of obese subjects. These aspects are considered in a global perspective in two populations in which MHO and MUO subjects were matched by (body mass index) BMI, sex and age.

## 2. Materials and Methods

### 2.1. Study Populations

Two populations were included in this cross-sectional study. Population 1 consisted of participants in the Obesity, Nutrigenetics, Timing, Mediterranean study (ONTIME), attending clinical centers to lose weight (clinicaltrials.gov: NCT02829619). Participants were from the province of Murcia, Spain, and were recruited from January 2008 to August 2019. From the total sample of 4543 subjects, those who had a BMI ≥25 kg/m^2^ and whose analysis of biochemical parameters was complete (glucose, HDL-cholesterol (high density lipoprotein-cholesterol), triglycerides, and insulin) were selected. Exclusion criteria were subjects diagnosed with diabetes and the use of drugs for losing weight, and glucose-lowering, lipid-lowering and antihypertensive treatments. After inclusion and exclusion criteria were considered, a total of 2634 participants were included in the matching analysis. The chosen subjects were matched for age (criterion ±2 years), sex and BMI (criterion ±2 kg/m^2^), considering a 1:1 ratio of MHO and MUO subjects; using MedCalc software v.15.0 (MedCalc Software, Ostend, Belgium), which provided 429 participants per group of MHO and MUO. Eighty percent (*n* = 346) of MHO participants and 82% (*n* = 353) of MUO had a BMI ≥30 kg/m^2^. All subjects signed an informed consent document. The study was approved by the Ethics Committee of Murcia University (ID: 1632/2017).

Population 2 comprised patients (*n* = 72) from “Virgen de la Arrixaca” and “Morales Meseguer” hospitals, who had undergone laparoscopy or abdominal surgery for reasons that did not interfere with the study and who also signed the informed consent document. Nineteen participants were MHO and 53 were MUO. From these subjects, 63% (*n* = 12) of MHO and 79% (*n* = 42) of MUO subjects had obesity. The same inclusion and exclusion criteria as used for Population 1 were considered. The study was approved by the Ethics Committee of “Virgen de la Arrixaca Hospital”. This study was conducted according to the principles expressed in accordance with the Helsinki Declaration [17].

### 2.2. MHO Characterization

According to the cut-off points of the National Cholesterol Education Program Adult Treatment Panel III [18] and the cut-off point for insulin resistance described by Matthews et al. [19], MHO consisted of those subjects with a BMI ≥25 kg/m^2^ who had alterations in one or none of the following parameters: triglycerides ≥150 mg/dL, HDL-cholesterol <40 mg/dL in men or <50 mg/dL in women, fasting glucose ≥100 mg/dL and HOMA-IR (homeostatic model assessment-insulin resistance) >2.5. Participants with a BMI ≥25 kg/m^2^ and two or more altered metabolic parameters were considered as MUO [20,21].

### 2.3. Anthropometric Measurements and Body Composition

In populations 1 and 2, body weight was measured using digital scales (Tanita Corporation of America, Arlington Heights, IL, USA) to the nearest 0.1 kg without shoes, wearing light clothes and after 12 h of fasting. Height was measured with a Harpenden digital stadiometer (Holtain Ltd. Crosswell, Crymyh, Pembs, UK). Waist circumference was assessed at the level of the umbilicus, and hip circumference overlying the greater trochanter of the femur. In Population 1, body fat percentage was assessed by means of bioelectrical impedance, using TANITA TBF-300 equipment (Tanita Corporation of America, Arlington Heights, IL, USA). In Population 2, body fat percentage was calculated with the Siri equation, using the skinfolds (biceps, triceps, suprailiac, and subscapularis), as previously described [22].

#### Computed Tomography (Population 2)

Abdominal subcutaneous and visceral adipose tissue (VAT) were evaluated by computed tomography according to the technique of Sjöstrom [23], using a Toshiba scanner, model CBTB007A (Toshiba Corporation 1385-1, Shimoshiqui, Otowawara, Japan), at the level of L4-L5 (lumbar 4-lumbar 5) with a 512 × 512 matrix, a window width of 300 Hounsfield units and a center of 40 Hounsfield units [22].

### 2.4. Lifestyle Factors (Population 1)

#### 2.4.1. Composition of Food Intake and Eating Behavior

Dietary intake was assessed through a 24 h recall. Energy intake and macro- and micronutrient composition were analyzed with the Nutrilet software v1.0 (Nutrilet, Murcia, Spain) based on Spanish food composition tables [24]. Eating behaviors were assessed by means of two questionnaires. The “Mediterranean Diet Score” [25] consists of nine items: vegetables, legumes, fruits and nuts, dairy, cereals and potatoes, meat and meat products, fish, monounsaturated/saturated FA ratio, and alcohol. The score goes from 0 (low quality diet) to 9 (high quality diet). The “Barriers to weight loss” questionnaire consists of 7 sections: meal recording, weight control, eating habits, portion size, food and drink choices, eating behaviors and other barriers [26]. Number of meals, timing and number of food groups portions were assessed through a nutritional record.

#### 2.4.2. Physical Activity

The International Physical Activity Questionnaire (IPAQ) [27], which has been validated for use in the Spanish population, was used to evaluate physical activity during the 7 days prior to enrollment. Briefly, the IPAQ questionnaire includes 7 items to evaluate three categories of physical activity: vigorous, moderate, and walking. For each category, frequency (days) and duration (minutes) of the activities is assessed. Then METs (metabolic equivalents) are calculated as follows: (a) walking METS = 3.3 × walking minutes × walking days; (b) moderate METS = 4.0 × moderate-intensity activity minutes × moderate days; (c) vigorous METS = 8.0 × vigorous-intensity activity minutes × vigorous days. Finally, total physical activity METs are calculated as the sum of walking METs + moderate METs + vigorous METs.

#### 2.4.3. Sleep Timing and Individual Chronotype

Based on our previous findings [28], we also included other factors that are involved in obesity such as sleep characteristics and individual chronotype. Sleep timing and duration were estimated by the following questions: “On weekdays (or weekends), at what time do you usually go to bed?” and “On weekdays (or weekends), at what time do you usually get up in the morning?”. Sleep duration was determined as the difference between bed and wake times. As no participants worked irregular shifts, weighted weekly sleep duration was calculated as: ((weekday sleep duration × 5) + (weekend sleep duration × 2))/7. Chronotype, sleep, and exercise timings were determined using the “Horne and Ostberg Morningness-Eveningness questionnaire”, which consists of 19 questions [29], the answers to which were dichotomized based on the median score of all the participants for easier interpretation.

#### 2.4.4. Other Lifestyle Factors

Several parameters were obtained such as smoking, obesity history of the patient, weight at birth and psychological profile, which included questions about emotional eating.

### 2.5. Adipose Tissue Characteristics (Population 2)

#### 2.5.1. Morphometric Characteristics

Subcutaneous adipocytes were obtained from the periumbilical region, and intra-abdominal samples were from the perivisceral fat surrounding the gall bladder and omental fat. Two samples were obtained from each adipose zone; one to study the fatty acid composition and the other to calculate the size of adipocytes. Adipocytes size was determined with the technique proposed by Sjöstrom et al. [23]. Briefly, with a microscope (Olympus CoII, Barcelona, Spain), using an ocular micrometer divided into 100 parts of 7.5 µ each, 200 cell diameters were measured with a magnification of 200×. To eliminate possible errors caused by the polarization of fat cells, 50 diameters were measured in one direction and 50 perpendicular to the first. Cells were measured throughout the thickness of the sample. Adipocytes sizes were estimated from their maximum diameter and the volume was calculated assuming that cells were spheres. Cell weight was determined from the volume considering the density of the triglycerides of the fat cell (0.915 g/mL). The Sjöstrom technique avoids the changes in size that adipocytes can undergo with traditional staining techniques and it measures thicknesses always greater than the average diameter of the adipocyte, which allows the measurement to be calculated from the entire adipocyte, as if it were a sphere seen from above. Fat cell size variability was determined using the standard deviation. All measurements were made in duplicate by the same operator. Nevertheless, to evaluate intra-operator variability, measurements were performed in duplicate in several subjects (*n* = 7), in the same histological section, and, in two different sections of the same sample, and compared with measurements performed by another operator. The correlation factors were r = 0.99, r = 0.99, and r = 0.94, respectively. Adipocyte numbers were calculated by dividing total body fat in kg by the average of the mean fat cell weights of the subcutaneous, perivisceral and omental regions [30].

#### 2.5.2. Fatty Acid Composition

To determine the FA composition of AT, total lipids from AT were extracted according to the method of Folch et al. [31]. FA methyl esters were identified on a Perkin-Elmer 84–10 gas chromatograph (Perkin-Elmer, Norwalk, CT, USA) equipped with a 30 m × 0.25 mm fused-silica capillary column (SP-2380, Teknokroma, Barcelona, Spain). The oven temperature was programmed for 7 min at an initial temperature of 160 °C and was increased at a rate of 3 °C/min to 230 °C and maintained at that temperature for 7 min. The injector and detector were set at 210 and 280 °C, respectively. Nitrogen was used as the carrier gas with a flow of 50 cm/s. The instrument output was quantified with a Perkin-Elmer GP-100 integrator (Perkin-Elmer, Norwalk, CT, USA). Peaks were identified by comparison with standard (Supelco, Bellefonte, PA, USA) [31].

### 2.6. Biochemical Determinations

For both populations, biochemical determinations were made after 12 h of fasting. Serum glucose, total cholesterol, HDL cholesterol, triglycerides, uric acid, and urea concentrations were determined with a BA400 biochemical analyzer (Biosystems, Barcelona, Spain). LDL-C (low density lipoprotein-cholesterol) was calculated using Friedewald’s formula when triglycerides were <400 mg/dL (i.e., total cholesterol minus HDL-C minus triglycerides/5) [32].

Serum insulin was measured with a solid-phase, enzyme-labeled chemiluminescent immunometric assay (IMMULITE 2000, Siemens Healthcare, Madrid, Spain). HOMA-IR was calculated as follows: (insulin (μUI/mL) × glucose (mg/dL))/405 [19]. In Population 2, 17-β-estradiol, testosterone, dehydroepiandrosterone sulfate (DHEA-S), sex hormone binding globulin (SHBG), androstenedione, c-peptide, tumor necrosis factor-α (TNF-α), and leptin were measured as previously reported [22].

### 2.7. Statistical Analysis

Quantitative variables that compare MHO and MUO phenotypes are presented as estimated mean ± standard error of the mean (SEM) (when data were adjusted in ANCOVA (analysis of covariance)) or mean ± standard deviation (SD) (when not adjusted) for Population 1 and 2, respectively. The proportion of men and women were expressed as percentages and compared with Chi-square test. BMI and uric acid correlations were performed with Pearson’s test. In Population 1, ANCOVA analysis was carried out, with the study number and clinical center as covariates, and a Bonferroni test was used for multiple comparisons. Logistic regression was used to estimate the odds ratios and the 95% confidence intervals of behavioral data, adjusted for study number and clinical center to compare MHO and MUO phenotypes. In Population 2, there were no differences in age, sex, or BMI between the MHO and MUO groups, and so the analyses were not adjusted for these variables; a Student t test or Mann–Whitney U test was used to compare quantitative variables as appropriate. Based on the previous results [33], and before starting the study, sample size was calculated for Population 2, and a minimum of 17 subjects per group were estimated to have enough power (80%) to detect a significant difference in VAT between MHO and MUO for alpha =0.05. To classify subjects as hypertrophic or hyperplasic obese, total body fat in kg was log transformed and plotted against fat cell volume by simple linear regression. Subjects above the regression line were considered as hypertrophic and those below the line as hyperplasic [30]. Due to incomplete information, we could only classify 9 MHO and 29 MUO subjects as hypertrophic or hyperplasic subcutaneous adipocytes and 9 MHO and 20 MUO subjects as hypertrophic or hyperplasic perivisceral adipocytes. Statistical analyses were conducted using SPSS v.20.0 software (IBM Corp., Armonk, NY, USA), and a *p*-value of < 0.05 was considered statistically significant. Figures were made with GraphPad v8.0.0 (GraphPad Software, San Diego, CA, USA). Graphical abstract was created using Biorender.com [34].

## 3. Results

### 3.1. Anthropometric and Body Composition Characteristics of MHO and MUO Subjects

Populations 1 and 2 had similar proportions of men and women. MHO subjects had a higher hip circumference and lower waist to hip ratio than MUO (Table 1 and Table 2). In Population 2, MHO had a lower visceral area than MUO, while no differences were found in subcutaneous area between the two metabolically obese phenotypes (Figure 1A,B).

### 3.2. Lifestyle Factors

Analysis of energy intake, macronutrients, and micronutrients showed no differences between MHO and MUO subjects, except for iodine and potassium intake, the consumption of both micronutrients being higher in MHO participants (Appendix A: Energy and nutrient intake in MHO and MUO (Population 1)). The Mediterranean Diet Score was similar for both metabolic phenotypes (Table 3). As regards eating behavior, MHO subjects usually had a complete breakfast (1.59× more likely in MHO than in MUO), a lower tendency to overeat when stressed (1.96× less likely than MUO), good adherence to dietary rules (1.48× more likely than MUO), with less snacking (2.09× less likely than MUO), a higher number of meals and higher consumption of fruit portions than MUO individuals. In contrast, the time of food intake was similar between phenotypes (Table 3).

No differences were found as regards physical activity, sleep timing, or total chronotype score; however, MHO presented characteristics of morning chronotype. MHO subjects tended to exercise in the morning rather than evening (physical exercise in the morning was encountered 1.54× more likely in MHO group than in MUO) and to go to bed earlier (Table 3). An earlier bedtime (8:30–10:30 p.m.) was associated with double the propensity to have an MHO phenotype compared with subjects who usually went to bed between 1:00–3:00 a.m. Furthermore, non-obesity during infancy was 2.07× more likely in MHO than in MUO, and smoking was encountered 1.74× less likely in the MHO group than in MUO (Table 3).

### 3.3. Adipose Tissue Characteristics (Population 2)

#### 3.3.1. Morphology

MHO subjects had smaller adipocytes and a more homogenous AT in the perivisceral/omental depots than MUO participants (Figure 1C,D), while no significant differences were found in the number of adipocytes between the two metabolically phenotypes (Figure 1E). The frequency of hypertrophic adipocytes was lower among MHO, although not to a statistically significant extent (Figure 1G,H).

#### 3.3.2. Fatty Acid Composition

When the FA composition of body fat was analyzed, it was observed that subcutaneous fat in MHO individuals had higher concentrations of 18:3 n-6 gamma linolenic fatty acid (GLA) and 20:5 n-3 eicosapentaenoic fatty acid (EPA) than MUO (Figure 1F). No significant differences were observed in other FA types (Appendix A: Fatty acids composition in subcutaneous, perivisceral, and omental adipocytes (Population 2)).

### 3.4. Biochemical Parameters

Pearson correlation analyses demonstrated that, as BMI increases in Population 1, so does the concentration of serum uric acid in both MHO and MUO, although it was always lower in MHO than in MUO for any BMI value (Figure 2A). In Population 2, a higher SHBG concentration was observed in MHO (Figure 2B) and the same trend was found in males and post-menopausal females separately (Appendix A: Hormonal concentrations (Population 2)). The levels of other hormones and proteins were similar for both phenotypes (Appendix A: Hormonal concentrations (Population 2)).

## 4. Discussion

In our study, we found significant differences between MHO and MUO subjects, such as timing of physical activity and sleep timing, both of which tended to be earlier in MHO subjects (more morning chronotype). Furthermore, the FA composition in AT was enriched in EPA and GLA. Other aspects such as eating behaviors, obesity status during infancy, adipocyte size, and distribution, biochemical, and anthropometric variables were also different between MHO and MUO (Figure 3).

MUO tended to exercise in the evening rather than morning. More importantly, later bedtimes (1:00–3:00 a.m.), a characteristic of evening chronotypes, were associated with double the risk of having an MUO phenotype compared to subjects who went to bed between 8:30 and 10:30 p.m. Both these factors are related to a more evening chronotype. Individual chronotype has been defined as a characteristic that determines an individual’s circadian preference [35]. Those individuals who prefer the first half of the day to carry out their activities are referred to as morning-types, and those who prefer the second part of the day are evening-types [35]. It has been observed that evening-types usually have a higher BMI [36], skip breakfast more frequently [8], and have a tendency to have greater visceral fat area than morning types [37], all of which, taken together, contribute to a poorer metabolic profile, as occurs in MUO subjects.

Moreover, MHO subjects had different dietary patterns to their MUO counterparts, mostly adhering to dietary rules, eating a complete breakfast, consuming more fruit, eating larger numbers of meals, with less snacking and lower tendency to overeat when stressed. These results closely reflect those of previous studies, which demonstrated that skipping breakfast is associated with an increased glycemic response in the postprandial state and prolonged exposure to free FAs during fasting and after lunch. In turn, this would diminish insulin sensitivity [38,39], as occurs in the MUO phenotype. Other authors have shown that eating six times per day reduces fasting insulin levels compared with eating three times per day [40], and a lower number of daily eating episodes has been associated with abdominal obesity and high levels of triglycerides [41]. On the other hand, a greater number of daily portions of fruit was associated with a lower risk of developing MUO phenotype [4,42]. These habits, together with less smoking, promote a healthier lifestyle among MHO individuals. By contrast, no significant differences were found in food intake in terms of total energy, macronutrient and micronutrient intake (except for iodine and potassium). Previous studies found similar total energy intake and dietary macronutrient intake in MHO and MUO individuals [4,43], which underlines that for maintaining a healthy metabolic status, dietary patterns and dietary quality seem to be more important than total energy intake and macronutrient distribution.

Obesity during infancy was seen to be another important lifestyle factor associated with MUO in this study. It has been observed that rapid or extremely rapid weight gain in the first fourth months of life, increases the risk of obesity by 50% at 2–7 years [44]; A U-shaped curve in body weight at birth (very low and very high weights) has also been associated with insulin resistance [45].

With respect to AT characteristics, such as FA composition, the subcutaneous adipocytes of MHO subjects had higher EPA and GLA concentrations than MUO. Other studies have investigated the FA profile, including EPA and GLA, in both metabolic phenotypes, but only in plasma, where no differences were found [12]. Previous results of our group have also shown that higher proportions of n-3 and n-6 FA from a various sources (including subcutaneous AT) are related with a lower metabolic risk [46]. Furthermore, a higher amount of n-3 and n-6 FA in AT was related with smaller adipocytes [47], while a higher amount of EPA in subcutaneous AT tissue lowered the expression of inflammatory genes [48]. Studies have shown that GLA suppresses acute and chronic inflammation [49]. Moreover, in an experimental model, a combination of EPA and GLA reduced acute inflammation through different pathways [50]. Inflammation is characterized by an altered secretion of molecules such as IL-6 (interleukin-6), IL-1β (interleukin-1β), resistin, MCP-1 (monocyte chemoattractant protein-1), and recently described IL-8 (interleukin-8), WISP1 (Wnt1-inducible signaling pathway protein-1), DDP4 (dipeptidyl peptidase-4), angio-poietin 2, chemerin, haptoglobin, and VAP-1 (vascular adhesion protein-1). This alteration has been associated with adipose tissue dysfunction, leading to the development of metabolic disorders [51]. Nevertheless, it has been reported that lipid mediators derived from n-3 FA suppress the activity of NFκB (nuclear factor kappa B) reducing the production of proinflammatory cytokines [52]. Future studies are needed to clarify whether high levels of EPA and GLA are associated with a lower inflammatory profile in MHO subjects.

As demonstrated in several previous studies, MHO subjects had a lower waist circumference and WHR (waist to hip ratio), and less visceral AT, as determined by computed tomography, than MUO subjects, which may partly explain the healthier metabolic profile that characterizes MHO subjects [53,54]. Other studies using computed tomography have been performed in postmenopausal women, though they consider different criteria to define MHO [33,54,55]; whatever the case, the lower visceral AT levels in MHO seem to be replicated in all the studies.

In addition, MHO subjects had smaller omental adipocytes than MUO subjects, which agrees with the findings of other studies [15,16]. In fact, large adipocytes have been described as tending to be dysfunctional and as secreting more proinflammatory cytokines [55]. They also tend to have reduced vascularization, which would lead to hypoxia and macrophage infiltration [56]. Larger adipocytes have a lower ability to suppress lipolysis [57], a factor that contributes to insulin resistance and metabolic alterations. Furthermore, visceral AT was more homogenous (similar size among cells) in the MHO than in the MUO phenotype. Accordingly, the frequency of hypertrophic obese subjects tended to be higher among MUO phenotype. Nevertheless, a small number of observations were analyzed, and results should be taken with caution and studies with larger sample sizes are needed.

SHBG is another biomarker that differed between metabolic phenotypes in the current study. Previous results have been shown to be contradictory [58,59]. Kavanagh et al. observed that obese women in the highest quartile of SHBG had the lowest visceral fat area [59]. Other authors reported hypertriglyceridemia and insulin resistance in subjects in the lowest quartile of SHBG [60,61]. A possible explanation of this relation between a higher SHBG and better metabolic profile is that higher concentrations of SHBG are related to higher lipoprotein lipase activity, the enzyme that catalyzes the hydrolysis of triglycerides of chylomicrons and VLDL-C (very low density lipoprotein cholesterol) [62].

As obesity has been positively associated with uric acid concentrations and higher levels of uric acid have been related to cardiovascular diseases [63], we analyzed the potential correlation of BMI and uric acid in MHO and MUO. Uric acid concentrations were found to be lower in MHO than in MUO and, regardless of their BMI, subjects with the MHO phenotype always had lower uric acid concentration. Most studies of MHO and serum uric acid have been performed in adolescents, with similar results [64,65]. Moreover, uric acid has been proposed as a biomarker to distinguish MHO and MUO subjects [63].

The strengths of this study are related to its rigorous evaluation of several lifestyle factors, including individual chronotype, sleeping, and eating behaviors. Furthermore, the strictly matched population permitted some confounding variables between the two metabolic phenotypes to be eliminated. Other strengths were access to AT samples, which permitted us to analyze FA acid composition and morphological characteristics. Among the limitations of this study are its transversal design and our lack of more objective instruments, such as accelerometers, to measure physical activity. Regarding adipocytes, we only measured morphometric characteristics and fatty acids composition, but it was not possible to measure other relevant biomarkers of adipose tissue function, such as adipokines, markers of hypoxia, apoptosis, inflammation, or fibrosis. Further research in the metabolomics field is needed, including information about lifestyle factors and the morphological characteristics of adipose, muscle and liver tissues, to clarify the heterogeneity and causes of different obesity phenotypes.

## 5. Conclusions

This study has shown that healthier eating habits, not smoking, and earlier sleep and exercise times could be recommended for improving the metabolic profile of subjects with obesity.

## Figures and Tables

**Figure 1 nutrients-13-00119-f001:**
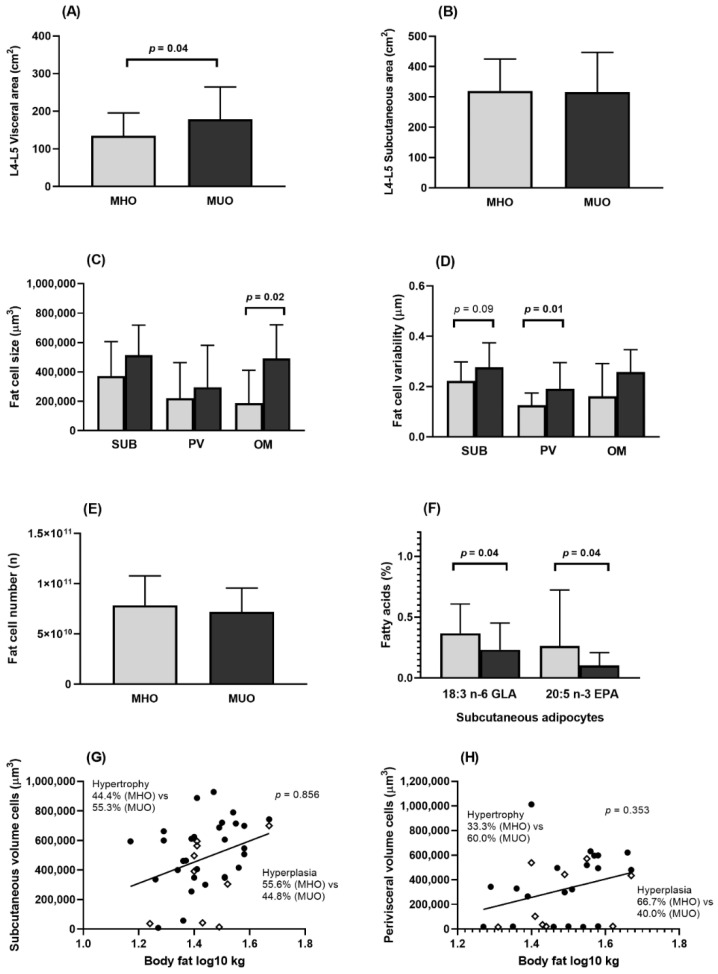
Body fat distribution and adipose tissue characteristics in MHO and MUO (Population 2). EPA, eicosapentaenoic acid; GLA, gamma linolenic acid; L4-L5, lumbar 4-lumbar 5; MHO, metabolically healthy obesity; MUO, metabolically unhealthy obesity; OM, omental; PV, perivisceral; SUB, subcutaneous. (**A**) Visceral area at the level of L4–L5 in MHO and MUO subjects. (**B**) Subcutaneous area (SUB) at the level of L4-L5 in MHO and MUO subjects. (**C**) SUB, PV, and OM adipocyte size in MHO and MUO subjects. (**D**) SUB, PV, and OM adipocyte size variability in MHO and MUO subjects. (**E**) Number of adipocytes in MHO and MUO subjects. (**F**) Fatty acid composition in SUB adipocytes in MHO and MUO subjects. (**G**) Proportion of hypertrophy and hyperplasia SUB adipocytes in MHO and MUO subjects. (**H**) Proportion of hypertrophy and hyperplasia PV adipocytes in MHO and MUO subjects. Figure 1A–F comparisons were made by Student t test or Mann–Whitney U test as appropriate. For (**G**,**H**), plots above the regression line have hypertrophy and those below the regression line have hyperplasia; comparisons of hypertrophy and hyperplasia proportions in MHO and MUO were made with Chi-Square test. Grey bars indicate MHO and black bars indicate MUO subjects. ◊ indicates MHO subject; ^●^ indicates MUO subject.

**Figure 2 nutrients-13-00119-f002:**
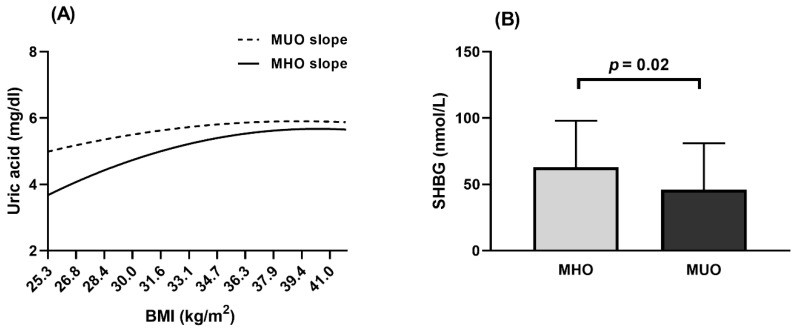
Biochemical parameters in MHO and MUO phenotype. (**A**) Population 1. Slopes of uric acid in MHO and MUO calculated by Pearson correlation. (**B**) Population 2. Comparison of SHBG concentration between MHO and MUO was made with ANCOVA analysis and adjusted for Study Number and Clinical Center. Grey bar indicates MHO and black bar indicates MUO subjects. BMI, body mass index; MHO, metabolically healthy obesity; MUO, metabolically unhealthy obesity; SHBG, sex hormone binding globulin.

**Figure 3 nutrients-13-00119-f003:**
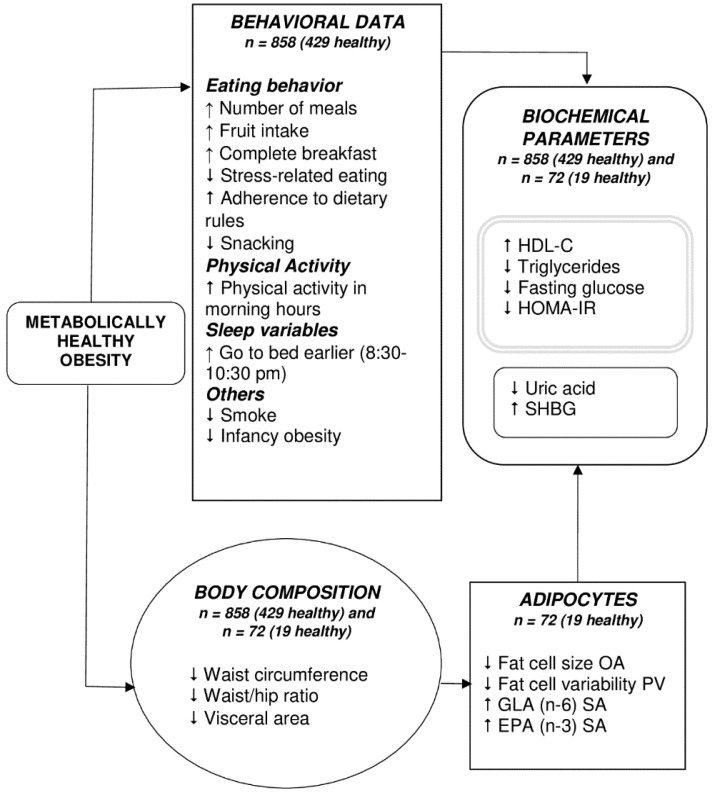
Potential mechanisms involved in Metabolically Healthy Obesity. ↑, more; ↓, less; EPA, eicosa-pentaenoic acid; GLA, gamma linolenic acid; OA, omental adipocytes; PV, perivisceral adipocytes; SA, subcutaneous adipocytes; SHBG, sex hormone binding globulin. The grey double-lined box includes those parameters used in the definition of MHO and MUO. The MHO phenotype was characterized by differences in lifestyle behaviors, adipocyte morphology and composition, body composition, and biochemical parameters, all of which probably interact to favor this phenotype.

**Table 1 nutrients-13-00119-t001:** General characteristics of Population 1.

Variables	MHO *n* = 429	MUO *n* = 429	*p*-Value
Anthropometrics and clinical parameters			
Sex (% F) ^1^	64.60	64.60	1.000
Age (y)	44.33 ± 0.63	44.84 ± 0.62	0.569
BMI (kg/m^2^)	34.52 ± 0.23	34.69 ± 0.23	0.587
Height (m)	1.65 ± 0.01	1.66 ± 0.01	0.311
Weight (kg)	94.33 ± 0.83	95.63 ± 0.82	0.265
BFP (%)	38.56 ± 0.34	39.35 ± 0.34	0.101
Waist (cm)	110.30 ± 0.62	112.89 ± 0.61	**0.003**
Hip (cm)	118.84 ± 0.48	117.50 ± 0.47	**0.046**
WHR	0.93 ± 0.01	0.96 ± 0.01	**<0.001**
Parameters used in the definition of MHO and MUO			
Triglycerides (mg/dL)	97.80 ± 3.01	169.10 ± 2.96	**<0.001**
HDL-C (mg/dL)	55.66 ± 0.64	45.96 ± 0.63	**<0.001**
Glucose (mg/dL)	85.17 ± 1.02	102.94 ± 1.00	**<0.001**
Insulin (µUI/mL)	7.18 ± 0.38	14.02 ± 0.37	**<0.001**
HOMA-IR	1.52 ± 0.11	3.61 ± 0.11	**<0.001**
Other biochemical parameters			
Total cholesterol (mg/dL)	197.06 ± 2.00	198.48 ± 1.97	0.613
LDL-C (mg/dL)	122.01 ± 1.72	119.89 ± 1.70	0.380
Uric acid (mg/dL)	5.19 ± 0.08	5.72 ± 0.08	**<0.001**
Urea (mg/dL)	39.90 ± 0.87	39.05 ± 0.86	0.491

Data are shown as estimated mean ± SEM according to ANCOVA analysis, adjusted for Study number and Clinical Center. ^1^ Variable calculated with Chi-Square test. % F, percentage of females; ANCOVA, analysis of covariance; BFP, body fat percentage; BMI, body mass index; HDL-C, high density lipoprotein cholesterol; HOMA-IR, homeostatic model assessment-insulin resistance; LDL-C, low density lipoprotein cholesterol; MHO, metabolically healthy obesity; MUO, metabolically unhealthy obesity; SEM, standard error of the mean; WHR, waist to hip ratio. Bold numbers highlight statistical significance.

**Table 2 nutrients-13-00119-t002:** General characteristics of Population 2.

Variables	MHO *n* = 19	MUO *n* = 53	*p*-Value
Anthropometrics and clinical parameters			
Sex (% F) ^1^	68.40	64.20	0.787
Age (y)	54.84 ± 11.20	52.92 ± 14.48	0.603
BMI (kg/m^2^)	32.11 ± 3.65	32.83 ± 3.85	0.481
Height (m)	1.58 ± 0.09	1.59 ± 0.10	0.732
Weight (kg)	79.97 ± 12.12	82.68 ± 13.30	0.438
BFP (%)	36.49 ± 7.32	34.48 ± 8.83	0.451
Waist (cm)	106.67 ± 10.20	111.17 ± 11.58	0.147
Hip (cm)	110.61 ± 8.10	109.88 ± 10.80	0.792
WHR	0.96 ± 0.06	1.02 ± 0.10	**0.042**
VA/SA	0.49 ± 0.37	0.69 ± 0.45	0.088
SA/VA	2.97 ± 1.71	2.37 ± 1.99	0.254
Parameters used in the definition of MHO and MUO			
Triglycerides (mg/dL)	116.21 ± 40.90	187.09 ± 88.24	**<0.001**
HDL-C (mg/dL)	60.89 ± 14.69	43.00 ± 12.70	**<0.001**
Glucose (mg/dL)	86.84 ± 6.32	127.06 ± 67.10	**<0.001**
Insulin (µUI/mL)	11.52 ± 7.24	17.40 ± 10.45	**0.027**
HOMA-IR	2.47 ± 1.51	6.27 ± 7.91	**0.001**
Other biochemical parameters			
Total cholesterol (mg/dL)	218.16 ± 47.82	210.40 ± 47.20	0.542
LDL-C (mg/dL)	130.42 ± 37.02	127.74 ± 38.37	0.803
Uric acid (mg/dL)	4.87 ± 1.57	5.09 ± 1.40	0.573
Urea (mg/dL)	30.74 ± 8.04	32.34 ± 13.71	0.633

Data are shown as mean ± SD and differences were calculated according to Student t test or Mann-Whitney U test as appropriate. ^1^ Variable calculated with Chi-Square test. % F, percentage of females; BFP, body fat percentage; BMI, body mass index; HDL-C, high density lipoprotein cholesterol; HOMA-IR, homeostatic model assessment-insulin resistance; LDL-C, low density lipoprotein cholesterol; MHO, metabolically healthy obesity; MUO, metabolically unhealthy obesity; SA, subcutaneous area; SD, standard deviation; VA, visceral area; WHR, waist to hip ratio. Bold numbers highlight statistical significance.

**Table 3 nutrients-13-00119-t003:** Behavioral data associated with MHO phenotype (Population 1).

	MHO VS. MUO		
**Eating Behaviour**	**OR**	**95% (CI)**	***p*** **-Value**
Have a complete breakfast ^1^			
Yes	1.59	(1.07–2.36)	**0.023**
Without stress-related eating ^2^			
No	1.15	(0.70–1.87)	0.588
Sometimes	1.96	(1.12–3.43)	**0.018**
Adherence to dietary rules ^3^			
Yes	1.48	(1.03–2.15)	**0.037**
Snacking ^4^			
No	2.09	(1.10–3.99)	**0.024**
**Dietary Intake**	**Mean ± SEM**	**Mean ± SEM**	***p*-Value**
Number of meals	4.10 ± 0.05	3.93 ± 0.05	**0.008**
Food timing (hour)			
Breakfast	8.42 ± 0.06	8.50 ± 0.06	0.359
Lunch	14.56 ± 0.04	14.57 ± 0.03	0.865
Dinner	21.37 ± 0.04	21.37 ± 0.04	0.971
Number of food groups portions			
Bread	5.51 ± 0.21	5.90 ± 0.21	0.184
Fruits	1.52 ± 0.08	1.30 ± 0.07	**0.043**
Vegetables	1.84 ± 0.09	1.74 ± 0.09	0.464
Proteins	7.28 ± 0.21	7.24 ± 0.20	0.913
Milk	1.39 ± 0.06	1.32 ± 0.06	0.376
Fat	4.93 ± 0.16	4.80 ± 0.16	0.552
Extra calories	338 ± 20	340 ± 20	0.941
Mediterranean Diet Score	3.44 ± 0.09	3.48 ± 0.09	0.758
**Physical Activity**	**OR**	**95% (CI)**	***p*-Value**
Timing ^5^			
Morning	1.54	(1.09–2.18)	**0.014**
	**Mean ± SEM**	**Mean ± SEM**	***p*-Value**
**Level (METs)**	3866 ± 401	3493 ± 407	0.515
**Chronotype and Sleep Habits**	**OR**	**95% (CI)**	***p*-Value**
Bed timing ^6^			
Early (8:30–10:30 pm)	2.11	(1.02–4.36)	**0.043**
	**Mean ± SEM**	**Mean ± SEM**	***p*-Value**
**Chronotype Score**	51.30 ± 0.76	50.83 ± 0.78	0.664
**Others**	**OR**	**95% (CI)**	***p*-Value**
Smoking ^7^			
No	1.74	(1.21–2.51)	**0.003**
Psyche profile ^8^			
Non-bulimic impulse	1.07	(0.23–4.92)	0.930
Non-depressant	2.64	(0.71–9.86)	0.148
Non-anxious	0.73	(0.53–1.03)	0.071
No obesity ^9^			
Infancy	2.07	(1.12–3.83)	**0.020**
Childhood	1.39	(0.73–2.64)	0.315
Adolescence	1.16	(0.64–2.10)	0.636
Adulthood	1.45	(0.88–2.38)	0.145
	**Mean ± SEM**	**Mean ± SEM**	***p*-Value**
**Weight at Birth (kg)**	3.80 ± 0.17	3.81 ± 0.16	0.992

Quantitative variables are expressed as estimated mean ± SEM analyzed by ANCOVA analysis and adjusted for Study Number and Clinical Center. Qualitative variables are shown as OR (95% CI) for MHO with respect to MUO and analyzed by logistic regression adjusted for Study Number and Clinical Center. For this analysis, reference categories were as follows: ^1^ = No, ^2^ = Yes, ^3^ = No, ^4^ = Yes, ^5^ = Evening, ^6^ = Late (1:00–3:00 am), ^7^ = Yes, ^8^ = Normal, ^9^ = Never. ANCOVA, analysis of covariance; CI, confidence interval; METs, metabolic equivalents; MHO, metabolically healthy obesity; MUO, metabolically unhealthy obesity; OR, odds ratio; SEM, standard error of the mean. Bold numbers highlight statistical significance.

## Data Availability

The data presented in this study are available on request from the corresponding author.

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
