# Peer review of "Healthy Obese Subjects Differ in Chronotype, Sleep Habits, and Adipose Tissue Fatty Acid Composition from Their Non-Healthy Counterparts"

_nutrients, 2020, doi:10.3390/nu13010119_

Round 1

Reviewer 1 Report

The paper by Torres-Castillo et al.

In this paper in two study populations ‘obese’ subjects were assigned to either the MHO or the MUO group, based on their plasma biochemical parameters. Hereupon groups were matched for age, sex and BMI. One of these populations underwent surgery and hence fat biopsies were obtained for analysis.

The study aimed to study life style and dietary intake differences between the two groups of obese subjects discerned, and (if applicable) in the features of the fat biopsies collected.

The study was well designed and clear cut in its approach, although by definition a BMI >30 is designated and considered as being obese. Individuals with a BMI >25 are considered as overweight, so perhaps authors can explain this somewhat more. As authors claimed to have matched the two groups to reduce confounding: what percentage of the groups formed had a BMI>30? Was this a match between groups? Authors could explain on what basis the 429 subjects in each group were selected from the 2634 eligible (BMI>25 and plasma data complete). This is now unclear and might have been biased and not at random… It is hard to believe that based on the exclusion criteria mentioned one ends up with exactly 429 in each group, so please elaborate.

The data presented are presented well but a bit fuzzy (unclear table format), and sometimes both as (part of a) table and figure, in which case the figure presentation is redundant (e.g. Fig 1 A and B, resp. visceral and subcutaneous AT area). Perhaps authors can critically review and perhaps include some more of the data now in supplement. Next, authors should present data of population 1 and 2 in separate and more legible tables, as in pop.2 far less subjects are included and different parameters are scored/analysed than in pop.1.

The paper is not well written and does not read well, also due to many ‘typos’ and incorrect word use/choice. I tried to list the errors below, but the paper needs some editing and linguistic reviewing.

The authors do not well explain methods used: how was physical activity assessed?, VLDL assay is missing, morphometric analysis method is lacking (HE stained histology?),

Fig 1 G+H: From pop.2 with 19 (MHO) and 53 (MUO) subjects in each group, resp, only data of some 9 subjects in the MUO group are depicted, but from the MHO group all 19 are shown in fig 1H, but more than 19 in 1G…how is this possible? Also, 4 diamonds (à) are drawn above the regression line and 5 below…and yet the conclusion reads 55.3% hypertrophs in this MUO group. And the same in fig 1H. Please check and/or explain. Apparently it was (more) difficult to collect the biopsies in the MUO group? Authors could also discuss/relate the value of the conclusion drawn on hypertrophy vs. hyperplasia, given this small number of observations.

In case of Table 2 authors express effects found in a OR value, do authors summarize the data in an OR value for MHO in respect to MUO? Authors should explain more clearly in Methods or caption (e.g. physical exercise in the morning is encountered 1.54x more likely in MHO group than in MUO, it is 1.74x less likely a subject in the MHO group is a smoker compared to MUO, etc.).

Looking at Table S1, assuming this concerns the daily intake (please state so!), I was a bit puzzled: subjects have an intake of about 2000 kcal/day, of which some 42% are lipids. Using the Atwater factors (1 g fat=9 kcal), brings hence a daily intake of 93 g lipids. Now authors mention 13.6 g of this lipid moiety are trans-fats…This is almost 15%...very high! Please check..

Table S2: these mainly hormonal data are the mix from male and female subjects? This might explain the huge variation in the data…and renders them useless. Did authors consider to present the data by gender? Perhaps this yields a more meaningful set, were all female subjects post-menopausal? Looking at the age in Table 1 this might not be the case…and part of the reason of the variation.

Authors correlate BMI with uric acid plasma levels: why? Can authors discuss what uric acid is an indication for, why is it interesting at all in this respect?

Fig 3 is a nice summary of the findings but no way a mechanistic explanation. Please amend, particularly the ‘cross’ on the dotted line, what do authors mean by this? In the grey box all parameters not differing between the groups in pop.1+2 are mentioned…can also be omitted as not contributing to the phenotypes discerned among obese subjects.

Finally, looking at Table 1 again: are most MUO subjects diabetic? Can authors confirm this an elaborate on? Or show the distribution among MUO vs MHO: it might be an as yet not identified (but obvious, and very relevant) discriminator whether an obese subject is MHO or MUO!

Textual remarks/typos (by line numbering)

43: Recent studies, which

52: use this concept method

56-57: their/them, mention to what these words refer to…is it the FA?

59+62: contemplate consider

65: factors that favour

67: rigorously

72: to loose weight?

75: excluding criteria

100: described

113: consists

114/5: way of eating? Obstacles to weight loss?

121: medical record: elaborate please…

126: birth weight

135: would be more objective with two independent observers!

136: dividing?

136: how was mean fat cell weight assessed? What was used as density value (1.06 g/ml?). Add please.

139-146: use min (not minutes), what was the flow of the He gas (ml/min)?

152: how was the value for VLDL obtained, used in the Friedewald equation?

152/3: formula when with triglycerides

158: were measured

161: what decided you to use either SEM or SD? Explain the abbreviation SD…

201: energy intake!

Table 2: birth weight

226: significance

244/5: novel obesity factors? towards a more advanced pattern? Explain or omit.

255: favour

266: worse less favourable

269: ties fit?

271: exposure

272/3: three time per day

282: was another important…

283: amend: I sure hope all babies gain weight in the first four months…what do authors mean to say? A too high weight gain or catch-up growth?

292/3: what does ‘Even though’ refer to? What is the contradiction? Dito for ‘Moreover’…

305: blood flow

311: K.?

314: SHBG is not a hormone!

316: explain/elaborate on uric acid…

321: elimination

329: obesity therapies? To become obese?

Reviewer 2 Report

In this study, the authors investigated differences in lifestyle and body fat composition and quality, in terms of lipid content and adipocyte size of SAT and VAT in relation to the metabolic phenotype of obesity, as indicated by the presence or not of metabolic disease (MHO vs MUO). The most interesting point of this article is the investigation of non-classical lifestyle factors in relation with these two phenotypes. Within environmental parameters that they took into account, early time of sleep and exercise in the morning were two novel factors associated with metabolically healthy obesity. I found this article original and up to date; the study design is well constructed and the statistical procedures are correct.

My main points are:

  • How did the authors calculate the sample size for population 2? Did they perform a sample size calculation? And a power analysis? Pleas add it to the statistics section.
  • I would have expected the authors to investigate also the presence of signs of VAT inflammation, which is a recognized risk factor for metabolic disease in overweight and obesity (doi:10.1007/s40618-019-01052-3; doi:10.1002/dmrr.3358) as also indirectly acknowledged by the authors in lines 303-309.
  • However, they only evaluated markers of altered lipid metabolism, i.e. adipocyte size, and lipid composition. No information on local adipokines, markers of hypoxia, apoptosis, inflammation and fibrosis are provided. It would have improved remarkably the value of the manuscript. If they cannot explore any of these parameters, at least they should discuss this point in the discussion section .
  • How did the authors choose the novel lifestyle factors to investigate in relation to metabolic impairment in obesity?
  • What do they mean with “psyche profile”? Please detail it to avoid misinterpretation of this expression.
  • The manuscript need a deep revision by a native English speaker. Many typos and grammar mistakes must be corrected (i.e. Change “MUH” into “MUO” in the abstract)

Round 2

Reviewer 2 Report

The authors have successfully answered to the major points raised during my first revision.